# Inhibitory Control under Threat: The Role of Spontaneous Eye Blinks in Post-Traumatic Stress Disorder

**DOI:** 10.3390/brainsci7020016

**Published:** 2017-02-04

**Authors:** Mikael Rubin, Denise A. Hien, Dipanjana Das, Robert D. Melara

**Affiliations:** 1Department of Psychology, City College, City University of New York, New York, NY 10017, USA; mikael.r13@gmail.com (M.R.); dr.denise.hien@gmail.com (D.A.H.); ddas4425@gmail.com (D.D.); 2Department of Psychology, University of Texas at Austin, Austin, TX 78705, USA; 3Derner Institute of Advanced Psychological Studies, Adelphi University, South Avenue, Garden City, NY 11530, USA; 4Department of Psychological & Brain Sciences, University of California at Santa Barbara, Santa Barbara, CA 93106, USA

**Keywords:** post-traumatic stress disorder, spontaneous eye blink rate, trauma exposure, flanker interference, cognitive control, inhibitory control

## Abstract

This study is the first to explore spontaneous eye blink rate (sEBR) in individuals with post-traumatic stress disorder (PTSD). We investigated the connection between the magnitude of flanker interference in PTSD participants and sEBR during performance on a modified version of the Eriksen flanker task. As a peripheral measure of cognitive control and dopaminergic function, sEBR may illuminate the relationship between PTSD and executive function. Findings revealed a positive relationship between sEBR and flanker interference in participants diagnosed with PTSD, to both threat-related and neutral stimuli, whereas this relationship was negative in participants exposed to trauma but without PTSD and in healthy controls. Although our results are suggestive of sEBR as a potential physiological index of emotional management in PTSD, most of the correlations were not significant, indicating that further research with a larger sample is needed.

## 1. Introduction

Post-traumatic stress disorder (PTSD), affecting between 3.6% and 9.7% of the general population [1], is a traumatic stress-related disorder characterized by intrusive thoughts and memories of traumatic events, avoidance of thoughts and feelings associated with the trauma, emotional numbing, and an amplified vigilance and alertness to environmental triggers following exposure to a life-threatening stressor. Results of neuropsychological tests reveal compromise to executive control functions in PTSD patients, particularly working memory (ability to maintain recent experiences in memory) and inhibitory control (ability to suppress task-irrelevant memories and experiences) [2,3,4,5]. The aim of the current study was to investigate the relationship between working memory (as indexed by spontaneous eye blink rate; sEBR) and inhibitory control of attentional interference (as measured by behavioral performance on the temporal flanker paradigm) in (a) individuals with PTSD; (b) individuals with trauma exposure but no PTSD (TE); and (c) healthy controls (HCs) with neither trauma exposure nor PTSD. The relationships between sEBR and attentional interference in PTSD and TE participants may help clarify underlying attentional processes that are thought to result from trauma exposure.

### 1.1. Spontaneous Eye Blink Rate (sEBR)

Spontaneous eye blinks represent a broad index of information processing functions spanning attention and working memory and are distinct from both reflexive and voluntary eye blinks [6]. Neuroimaging analyses reveal the role of spontaneous eye blinks in modulating activity in oculomotor regions—including the frontal eye-fields and the supplementary eye-fields [7]—and also in higher-level cortical regions, including the parahippocampal gyrus and the medial frontal gyrus [8]. The parahippocampal gyrus is intimately involved in the updating of working memory to task-relevant stimuli [9,10,11], whereas the medial frontal cortex has been linked to executive control of task-irrelevant stimuli, most specifically during the monitoring and resolution of stimulus conflict in Stroop-like tasks [12]. Both working memory and executive control are compromised in PTSD [2,3,4,5]. sEBR also has been strongly associated with the dopaminergic system [13,14,15]: Pharmacological interventions reveal close ties between sEBR and D2 dopamine receptor functioning [13,14,15]. Researchers hence have used the eye blink rate as a non-invasive measure of dopamine functioning. Patients with disorders that involve hyperdopaminergic states—including schizophrenia, Tourette’s syndrome, and Huntington’s disease—show higher blink rates than healthy controls, whereas those with disorders that have hypodopaminergic states—such as Parkinson’s disease—show lower blink rates [16]. In non-disease states, sEBR serves as a gauge of cognitive load [14,15,16] independent of motor function.

In cognitive research, investigators have consistently found that high rates of spontaneous eye blinks during cognitive task performance are associated with enhanced perceptual sensitivity [17,18,19,20]. One speculative hypothesis is that sEBR gauges the buildup [21] or release [22] from working memory of task-related stimulus representations. In this view, enhanced perceptual sensitivity from high sEBRs might yield better inhibitory control of distractors once working memory is freed up [23]. This interpretation is supported by findings of reduced flanker [24] or Stroop [25] interference (i.e., greater inhibitory control of distractors) in healthy participants with greater sEBR. Inhibitory control entails increased resistance to distracting information during the performance of a cognitive task [26]. This includes both inhibitory action control and inhibitory interference control. Inhibitory control can be described by the ability to ignore the conflict between targets and distractors (flankers), by concentrating only on the target. In a standard flanker task this represents inhibitory interference control. Thus, decreased flanker interference indicates greater inhibitory control [27].

In the context of exposure to trauma, sEBR may offer insight into distinct cognitive strategies for handling threats, possibly leading to differential outcomes in inhibitory control to threats observed in PTSD and TE individuals. Functional neuroimaging of PTSD participants during the performance of cognitive control tasks reveals abnormal activity in the ventromedial prefrontal cortex, anterior cingulate cortex, amygdala, and insula [28,29,30,31,32]. Inefficient prefrontal control mechanisms, combined with hyperactive amydala sensitivity, may result in the reduced inhibitory control to threats observed in PTSD [28,30,33].

The relatively greater sensitivity to threats seen in individuals with PTSD (e.g., [34]) may be expressed as greater monitoring of threat-related cues during task performance. Here, sEBR could gauge the accumulation of threat material in working memory (i.e., the filling up of working memory), which would effectively undermine the inhibitory control of conflict between targets and distractors in PTSD. In this case, higher sEBR would predict poorer inhibitory control (i.e., more interference) with exposure to task-irrelevant threatening stimuli. In contrast, the sEBR of TE individuals to threat-related stimuli may reflect the adaptive unburdening of emotional material from working memory (i.e., the emptying out of working memory), thus enabling greater cognitive control of stimulus conflict. In this case, higher sEBR would predict enhanced inhibitory control (i.e., less interference) with exposure to task-irrelevant threatening stimuli (i.e., threat suppression) in TE.

### 1.2. The Present Study

We addressed two gaps in the literature concerning the relationship between the control of working memory and the psychopathology of stress and PTSD. First, we evaluated in patients with PTSD predictive associations between the monitoring or control of working memory, measured here as sEBR, and the magnitude of flanker interference, as an inverse measure of inhibitory control to distractors. Second, we evaluated whether traumatic stress exposure per se influences inhibitory control by examining whether participants with PTSD use control processes differently from those in each of the two comparison groups (TE and healthy controls).

The current study was part of a larger study to examine inhibitory control in PTSD [35]. We used a modified flanker conflict task [36] in which targets and distractors were separated temporally. Participants were asked to make judgments of line orientation while ignoring task-irrelevant lines and threatening or non-threatening images. We employed two sets of distracting images—a set of faces and a set of International Affective Picture System (IAPS) scenes—to probe the generality of task-irrelevant images in PTSD. We recorded behavioral performance and sEBRs across blocks of threatening or non-threatening trials. Although the relationship between sEBR and symptomology has been explored in other clinical populations, to the best of our knowledge our study is the first to examine the link between sEBR and inhibitory control in the context of PTSD.

## 2. Methods

### 2.1. Participants

Fifty individuals were recruited via online and print advertisements and paid to participate in the study. The nature of the procedures was explained fully, and informed consent was obtained from each participant; the Institutional Review Board of the City University of New York approved the protocol #298190. Following written consent, participants were given a breathalyzer/urine toxicology screen and completed a demographics survey and a life events checklist (LEC [37]). Trained clinicians administered the Structured Clinical Interview for the Diagnostic and Statistical Manual (SCID-IV [38]; DSM-IV). Inclusion criteria included: (1) physically healthy with no serious medical problems; (2) normal or corrected normal visual acuity; (3) aged 18–65; and (4) able to provide informed consent. Exclusion criteria included: (1) substance use disorder; (2) a major depressive episode; (3) suicidality; (4) current or history of psychosis or bipolar disorder, and (5) lack of fluency in English. Participants also were administered the Clinician-Administered PTSD Scale (CAPS [37]), and completed the Multiscale Dissociation Inventory (MDI [39]), the Test of Nonverbal Intelligence (TONI-4 [40]), and the Beck Depression Inventory^®^-II [41]. Three groups were created: (1) PTSD (*n* = 19, 10 males, mean age = 35.3), participants who met all DSM-IV-TR [42] criteria for PTSD; (2) TE (*n* = 16, eight males, mean age = 34.3), participants without PTSD or an Axis I diagnosis who had experienced a traumatic event meeting Criterion A for PTSD; and (3) HC (*n* = 15, five males, mean age = 33.4), participants without PTSD, an Axis I diagnosis, or Criterion A trauma exposure.

### 2.2. Stimulus, Apparatus, and Procedure

Each participant completed 24 blocks of experimental trials (and one or more blocks of practice trials) in a modified version of the visual flanker task [36] called the temporal flanker paradigm. The study was carried out in an electrically and acoustically shielded Industrial Acoustics Company (New York, NY, USA) chamber. Stimuli were created in Presentation^®^ version 16.0 software (Neurobehavioral Systems, Berkeley, CA, USA) on a Dimension 5150 Dell desktop computer and appeared to participants as they sat at a distance of 60 cm from a 17-inch Dell Model P1130 RGB computer monitor with a refresh rate of 75 Hz. Each task in the paradigm comprised a block of 80 trials. Each trial consisted of a fixation square (0.67°) followed by three stimulus displays presented sequentially: (1) First Flanker; (2) Target; and (3) Second Flanker (see Figure 1). Each display appeared for 150 ms separated by an inter-stimulus interval varying between 153 ms and 390 ms in random distribution. The first and second flankers were identical on each trial: a vertical line, a horizontal line, or a cross. Each target was a vertical or horizontal line superimposed on an image, either a grayscale threatening (i.e., fearful) or non-threatening face from the NimStim set [43] or a colored threatening or non-threatening picture from the International Affective Picture System (IAPS [44]). Line stimuli, subtending a visual angle of 0.57°, appeared in gray on a black background; face and IAPS stimuli subtended 9.93° (V) × 7.13° (H) and 10.85° (V) × 9.74° (H) of visual angle, respectively. We included two types of threat cues in the current study—faces and IAPS images—to evaluate the generality of the relationship between sEBR and flanker interference in PTSD when confronted with threatening versus non-threatening material.

On each trial of each task participants were asked to respond by mouse key as quickly and accurately as possible to the orientation of the target line, ignoring the flanker lines and the surrounding image. To mitigate eye movement artifacts during EEG recording participants were asked to blink only before or after, but not during, stimulus presentation, as needed. The duration of stimuli was 1265 ms on average within a trial that lasted approximately 3665 ms. Assignment of line orientation to response keys was counterbalanced across participants. Tasks were divided into baseline tasks (distractor constant) and filtering tasks (distractor random) [45]; for purposes of the present study, only performance on filtering tasks was analyzed. In each filtering task distractors appeared randomly on each trial: On 40% of trials (32 of 80) target and flanker lines matched in orientation (congruent trials), on 40% they mismatched (incongruent trials), and on 20% (16 of 80) flankers were crosses (neutral trials). All eight images from one of the four sets (threat faces, non-threat faces, threat IAPS, non-threat IAPS) appeared randomly an equal number of times with the target line, creating four different filtering tasks. Each task set was repeated three times in a session; task order was balanced across participants. The entire experiment, including EEG preparation, lasted approximately 3 h.

### 2.3. Data Recording and Analysis

Reaction times (RTs) to correct response trials of each participant were averaged for each of two trial types (i.e., congruent, incongruent) in each of four conditions (i.e., threat faces, non-threat faces, threat IAPS, non-threat IAPS). Our measure of inhibitory control (i.e., flanker interference) was defined as the difference between congruent and incongruent trials in each condition.

This study was part of a larger EEG study using a BioSemi Active-Two system in a high-density (160 electrodes) montage arranged in an elastic cap with blinks and other eye movements monitored by electrooculogram (EOG) at a sampling rate of 512 Hz from two electrode montages, one on the infra- and supra-orbital ridges of the right eye (VEOG), the other on the outer canthi of each eye (HEOG) (see [35]). For purposes of the present study, only VEOG recordings were analyzed. An automated Matlab function called Peakfinder [46] was used to identify and count blinks in the VEOG channel. Peakfinder implements a user-defined magnitude threshold to compare temporally adjacent peaks within a noisy unidimensional vector. Blinks were defined as peaks with a magnitude greater than 100 µV [47]. Blink frequency was summed across the three repetitions of each condition and divided by condition duration (in minutes) to determine average blink rate per minute (EBR) for each of the four conditions. The automated blink detector was checked for accuracy by two trained raters independently scoring a random selection of 30 blocks. No blinks were detected at a threshold lower than 100 µV (no false positives). The raters revealed 98% agreement in identifying blinks; the manual counts were 92% similar to the automated ones.

Two-tailed correlations were carried out within each of the three groups (i.e., PTSD, TE, HC) to examine the relationship between sEBR and inhibitory control in each condition. Additionally, we conducted exploratory analyses to examine whether specific clinical domains are related to sEBR. Correlational analyses were performed on clinical measures by separately regressing each subscale of the CAPS, BDI, and MDI on sEBR in the PTSD and TE groups. Fisher r-to-z transformations were performed to evaluate differences in the magnitude of correlations between groups (*p* < 0.05 for all tests).

## 3. Results

### 3.1. Demographic, Clinical, Physiological, and Behavioral Data

Table 1 contains a tally and statistical analysis of demographic and clinical data. PTSD and TE showed significantly higher rates of trauma than HC. PTSD revealed higher scores on the BDI than TE or HC. The three groups did not differ significantly in gender, age, education or race/ethnicity. The three groups did not differ significantly in sEBR or flanker interference.

### 3.2. Inhibitory Control

Table 2 summarizes correlation coefficients between sEBR and flanker interference scores (incongruent RT minus congruent RT) for each group in each of the four conditions. A pattern of correlations emerged in which sEBR and flanker interference consistently correlated positively in the PTSD group and consistently correlated negatively in the TE and HC groups. At an overall level (across conditions), the correlation coefficients were not statistically significant in two-tailed tests: *r* = 0.43, *p* = 0.067 (PTSD); *r* = −0.25, *p* = 0.37 (TE); *r* = −0.28, *p* = 0.30 (HC). However, there was a significant difference in the correlation coefficients between the PTSD and TE groups, *z* = −1.98, *p* < 0.05. Moreover, the PTSD group showed a significant positive association between sEBR and flanker interference in the threat IAPS condition, *r* = 0.51, *p* < 0.05, which differed significantly from the (nonsignificant) negative correlations between sEBR and flanker interference in the TE (*r* = −0.20; *z* = −2.29, *p* < 0.05) and HC (*r* = −0.29; *z* = −2.01, *p* < 0.05) groups.

### 3.3. Clinical Subscales

Exploratory two-tailed correlations were carried out in the PTSD and TE groups between sEBR and clinical subscales of the CAPS (four subscales) and MDI (six subscales), and the overall BDI. The blink rate correlated negatively only with the emotional constriction subscale of the MDI in the PTSD group, *r* = −0.48, *p* < 0.05. (The five items from the emotional constriction subscale are: “Not being able to feel emotions”; “Feeling frozen inside without feelings”; “Knowing you should be mad or sad about something, but not having any feelings”; “Not having emotions or feelings at a time when you should have been upset”; “Knowing you must be upset, but not being able to feel it”).

## 4. Discussion

The current study investigated links between the magnitude of inhibitory control in PTSD participants and their rate of eye blinks during performance of a temporal flanker task. We found that sEBR in participants diagnosed with PTSD was associated positively with flanker interference: Participants evinced a higher blink rate as the difference in RT between congruent and incongruent stimuli increased in the face of both threatening and non-threatening IAPS images. Thus, higher sEBR was associated with poorer attentional performance in PTSD. In contrast, in healthy control participants, as well as in participants who had been exposed during their lifetime to traumatic stressors but did not have PTSD, there was no significant association between blink rates and inhibitory control; at most, they exhibited a trend towards the usual relationship of sEBR with better attentional performance. Importantly, the correlations between PTSD and TE differed significantly from each other, suggesting that sEBR may help elucidate differences in underlying cognitive processes between the two groups.

In the PTSD group the IAPS stimuli yielded numerically higher correlation coefficients than the face stimuli (see Table 2). Interestingly, the difference between stimulus sets was particularly robust to threat stimuli. The effect cannot be understood merely as a weaker response in PTSD to facial processing because faces also appeared in many of the IAPS images. Perhaps IAPS images are more potent stimuli for eliciting threat because, in depicting threatening scenes, they are more directly threatening to PTSD participants, whereas, in expressing fear, NimStim faces depict only the aftermath of (i.e., the response to) threat. We believe this tentative explanation merits further investigation (see Shin and Liberzon [48]). In any case, we conclude that in our cue reactivity paradigm, the PTSD participants only weakly experienced the fearful faces as trauma cues compared with the IAPS images.

Hester and Garavan [23] hypothesized that sEBR predicts the success of cognitive processes responsible for inhibiting the influence of task-irrelevant stimuli on performance. At high rates of spontaneous blinking, the contents of working memory are dispersed [22], thus enabling frontal executive mechanisms to exert top-down control over distractors. Here, blinks are linked closely to cognitive control [24,25]. This perspective fits with our findings in the TE and HC groups. The negative correlations between sEBR and flanker interference in these groups suggest that greater sEBR is related to greater cognitive control. We speculate that there may be an emotional component to this process: Individuals in these two groups may be able to release threatening and neutral material equally well from working memory, enabling executive control of stimulus conflict across all incongruent trials.

A different understanding of the role of sEBR is desirable in explaining the performance of our participants with PTSD. Optimal levels of dopamine during the performance of conflict tasks [49] correspond to the peak of an inverted U-shaped function [50]. PTSD has been linked to abnormally elevated release of presynaptic dopamine [51,52] in both the amygdala [50] and dorsal anterior cingulate cortex (dACC) [53], implying inefficient inhibitory control of hypersensitive limbic regions [54]. To the extent that sEBR gauges hyperactivation in these regions, the positive correlation of the blink rate with flanker interference found in the current study may represent poor overall inhibitory control in PTSD.

What cognitive or clinical functions might be served by a positive correlation between sEBR and interference in PTSD? It is conceivable that sEBR acts as a check on the flow of information entering working memory [21]. A diagnostic feature of PTSD is heightened sensitivity to environmental threat [55,56]. On this account, blinks in individuals with PTSD may serve as a proxy for the threat monitoring processes that these patients engage in automatically to complete everyday tasks [57]. In particular, sEBR in this population could gauge the buildup of threat material within working memory, a process at cross-purposes with the cognitive control needed to ameliorate stimulus conflict. sEBR in those with PTSD may reflect the output of a hypersensitive emotion management system, rather than a cognitive control system. Emotional management involves both emotion monitoring, observed here in the positive correlation in PTSD between sEBR and flanker interference, and emotion control, observed in our exploratory findings indicating a negative correlation in PTSD between sEBR and the dissociative symptom of emotional constriction. Importantly, we found correlations of nearly equal magnitude to both threat and non-threat images. Responsivity under conditions of low threat in PTSD has also been reported elsewhere [58,59]. An explanation favored by previous investigators, and in concert with our emotional management account, is that in assessing risk patients with PTSD, vigilantly monitor both safety and danger. Still, at present our explanations are speculative and our results are in need of replication.

## 5. Conclusions

The current study is the first to explore sEBR in individuals with PTSD. We showed that whereas increasing rates of spontaneous eye blinks are associated negatively with flanker interference in TE and HC, sEBR and flanker interference are actually associated positively in PTSD. One implication is that sEBR represents the engagement of differential functions for individuals with and without PTSD, enabling either improved cognitive control (TE) or more effective emotion management (PTSD). To probe these differential uses, in future research sEBR could serve as the dependent measure in manipulations of working memory load with threatening or non-threatening material in participants from PTSD and TE populations.

## Figures and Tables

**Figure 1 brainsci-07-00016-f001:**
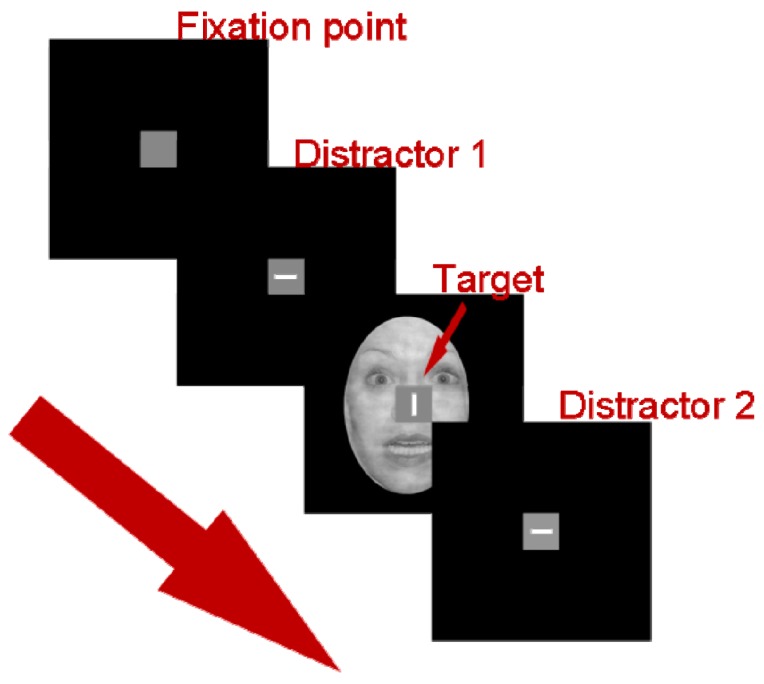
The modified flanker task. This example is of a fearful face incongruent trial.

**Table 1 brainsci-07-00016-t001:** Demographic, clinical, physiological, and behavioral information and comparisons among participants.

	PTSD (*n* = 19)	TE (*n* = 16)	HC (*n* = 15)	Test Statistic	*p*
Gender					
Male	10	8	5	χ^2^(2, *N* = 50) = 1.38	0.50
Female	9	8	10		
Age M (SD)	35.3 (10.1)	34.3 (12.0)	33.4 (9.7)	F(2, 47) = 0.129	0.88
Education (years) M (SD)	14.7 (1.9)	14.3 (1.8)	14.1 (4.7)	F(2, 47) = 0.151	0.86
Race/Ethnicity					
White	3	3	3	χ^2^(2, *N* = 50) = 2.61	0.27
Black	8	5	6		
Latino	6	3	4		
Asian	1	4	1		
Other	1	1	1		
Trauma type (LEC)					
Physical Abuse/Assault	16	11	3	χ^2^(2, *N* = 50) = 5.93	0.00
Sexual Trauma	17	11	1		
Natural Disaster	6	6	7		
War/Combat	2	2	2		
Other	36	44	24		
BDI	11.3 (7.9)	3.8 (5.4)	0.47 (0.92)	F(2, 47) = 15.8	0.00
Blink rate (mean) M (SD)					
Non-threat faces	15.9 (8.5)	20.8 (9.2)	19.4 (15.4)	F(2, 47) = 0.898	0.41
Threat faces	15.2 (9.1)	20.1 (8.5)	19.6 (12.2)	F(2, 47) = 1.292	0.28
Non-threat IAPS	17.3 (9.4)	19.4 (11.3)	17.6 (11.1)	F(2, 47) = 0.200	0.82
Threat IAPS	16.3 (8.4)	17.5 (12.2)	17.3 (10.1)	F(2, 47) = 0.066	0.94
Overall	16.3 (8.4)	19.2 (9.5)	18.2 (11.3)	F(2, 47) = 0.408	0.67
Flanker interference M (SD)					
Non-threat faces	112.1 (100.0)	133.1 (102.0)	111.8 (89.8)	F(2, 47) = 0.923	0.41
Threat faces	98.7 (86.3)	114.9 (110.4)	119.2 (108.3)	F(2, 47) = 0.999	0.38
Non-threat IAPS	119.4 (97.3)	142.5 (105.9)	97.8 (63.9)	F(2, 47) = 0.200	0.82
Threat IAPS	94.0 (83.7)	140.2 (129.7)	109.2 (69.2)	F(2, 47) = 0.254	0.78
Overall	106.0 (84.2)	132.7 (104.0)	109.4 (78.2)	F(2, 47) = 0.436	0.65

PTSD = Post-traumatic Stress Disorder, TE = Trauma Exposed, HC = Healthy Controls, LEC = Life Events Checklist, BDI = Beck Depression Inventory, IAPS = International Affective Picture System.

**Table 2 brainsci-07-00016-t002:** Correlation coefficients between sEBR and inhibitory control scores (incongruent RT minus congruent RT) for PTSD, TE, and healthy control participants in each of four conditions: threat faces, non-threat faces, threat IAPS, and non-threat IAPS.

Interference Condition	PTSD Blink Rate (*n* = 19)				TE Blink Rate (*n* = 16)				HC Blink Rate (*n* = 15)			
	NF	TF	NI	TI	NF	TF	NI	TI	NF	TF	NI	TI
Non-threat Faces	0.363				−0.225				−0.295			
Threat Faces		0.284				−0.201				−0.196		
Non-threat IAPS			0.415 ^†^				−0.219				−0.158	
Threat IAPS				0.510 *				−0.201				−0.286

PTSD = Post-traumatic Stress Disorder, TE = Trauma Exposed, HC = Healthy Controls, NF = Non-threat Faces, TF = Threat Faces, NI = Non-threat IAPS, TI = Threat IAPS; † *p* < 0.1, * *p* < 0.05.

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
