# Peer review of "Inhibitory Control under Threat: The Role of Spontaneous Eye Blinks in Post-Traumatic Stress Disorder"

_brainsci, 2017, doi:10.3390/brainsci7020016_

Round 1

Reviewer 1 Report

Hyperactivity in the amygdala was mentioned in the introduction. Some discussion of this topic could be included in the discussion of the current results. 

Author Response

POINT-BY-POINT RESPONSES TO REVIEWERS

Reviewer 1:

Hyperactivity in the amygdala was mentioned in the introduction. Some discussion of this topic could be included in the discussion of the current results. 

We agree, and have included discussion of amygdala hyperactivity in the Discussion section.

Reviewer 2 Report

Title: Inhibitory control under threat: The role of spontaneous eye blinks in post-traumatic stress disorder

Summary of Manuscript. The authors describe a study comparing participants with posttraumatic stress disorder (PTSD; n = 19), trauma-exposed controls without PTSD (TE; n = 16), and non trauma-exposed “healthy controls” without PTSD (HC; n = 15). All participants completed the temporal flanker paradigm. In this paradigm participants are presented with a target visual stimulus (a vertical or horizontal line on a square superimposed on one of four types of images: threat faces, non-threat faces, threat non-faces, and non-threat non-faces). Each target stimulus is “flanked” (before and after) by stimuli that are either congruent (vertical or horizontal), incongruent (horizontal or vertical), or neutral (a cross) with respect to the line on the target stimulus). Participants are instructed to identify the orientation of the target stimulus while ignoring the flanker stimuli and the background images. They are also instructed not to blink during the trials. Their vertical electrooculogram (EOG) while viewing the images and reaction time to correct response were assessed. Eye blink rate (EBR) per minute within each of the four stimulus conditions was determined using specialized blink detection software. Flanker interference (also referred to as inhibitory control) was determined by calculating the difference in average reaction time between the congruent and incongruent trials (incongruent-congruent) in each stimulus condition. The most notable finding was a pattern of positively signed correlations between EBR and flanker interference in the PTSD group but negatively signed correlations in the TE and HC groups. Only one of the 12 reported correlations was statistically significant, i.e., PTSD subjects showed a moderately strong positive association (r = .51) between EBR and flanker interference when presented with threat-related non-face slides. This correlation was significantly different from what was observed in the other two participant groups where the associations were small, negative, and not reliably different from zero. The authors also noted a significant negative correlation between EBR and a psychometic subscale indexing emotional constriction (r = -.48). They conclude that “EBR represents a potential physiological index of emotional management in PTSD.”

Overall Impression. This manuscript has several strengths. Foremost among these is the fact that it seems to be the first study to explore eyeblink rate in PTSD yielding a potentially interesting result. In addition, the study used established ways of assessing trauma, PTSD, and other psychopathology (i.e., LEC, CAPS, SCID-IV). Finally, the writing is generally clear and concise.

There are also significant weaknesses. Most notable to this reviewer is the unconvincing extrapolation from the observed data (i.e., eyeblink rate and flanker interference) to broader psychological constructs (i.e., working memory and inhibitory control). It is possible that the authors are correct in their interpretation of the psychobiological significance of their data but the manuscript does not provide sufficient, clear, proof of their claims. Absent that, the paper might be improved by restraining their inferences about the meaning of their findings (or at least acknowledging the speculation in their interpretations). One finding indicates that PTSD patients who blink excessively when confronted with a particular type of threat cue (i.e., non facial) also tend to be more susceptible to interference in the flanker task. There could be many reasons for the eye closures (beyond what is suggested by the authors) including attempting to avoid attending to a potential trauma cue (i.e., a core symptom of PTSD). Another finding indicates that PTSD patients who are elevated in a particular type of dissociation (i.e., emotional constriction) show slower blink rates. This latter finding could be interpreted in light of the dissociative subtype of PTSD as described in DSM-5. One wonders whether the finding applies to all of the experimental conditions particularly or whether it applies specifically to the aforementioned  “threat IAPS” condition? If so, would the study benefit from an analysis that accounted for these potential PTSD subtypes? Finally (as noted in detail below), the pattern of results in the PTSD group raises questions about why the IAPS images were more potent than the facial images. It suggests that this factor (rather than threat cue per se) may be relevant to interpreting the results. In sum, I am intrigued by the results but not entirely sold on the explanation for them.

Other Specific Suggestions for Improvement:

1.     EBR is described as “a broad index of information processing functions” (p. 2). This seems vague. Though space is limited, it would be helpful to have a concise description of the known neurophysiology of EBR to help the reader to understand its link with “information processing.” The information given about the dopaminergic underpinnings of EBR could suggest that it indexes motor function. Similarly, the authors state but do not explain links between EBR and “inhibitory control” in prior literature. Indeed the reader is told to expect that EBR would predict poorer inhibitory control in the PTSD group but better inhibitory control in the control groups. We are told that EBR reflects “accumulation of threat material in working memory” in the PTSD group but “adaptive unburdening of emotional material from working memory” in the control groups. This is confusing!  

2.     Explain why participants are “asked to blink before or after, but not during, trials as needed.” Does the study aim to partly evaluate the participant’s ability to control blinking in the face of threatening stimuli? If so, this could be more clearly stated.  In any case, this instruction raises concerns about characterizing the eyeblinks assessed in this study as “spontaneous.”

3.     Inhibitory control could be better defined.  Explain why more flanker interference indicates more inhibitory control to convince the reader that one can be substituted for the other.

4.     Justify the interest in contrasting faces and non-faces in the study design. The results from the two types of threat cues are not consistent.

5.     Table 1 would be improved by comparing the groups on mean levels of all dependent measures (i.e., EBR and flanker interference) for each stimulus condition. The interpretation of the correlations would be aided by knowing whether, for example, the PTSD group had a higher overall EBR (or only higher EBR under certain conditons).

6.     There is a discrepancy between the description of some results in the text (pg. 5) and Table 2. Specifically, the text reports “negative correlations between EBR and flanker interference in the TE (r = -.20; z = -2.29, p < .05) and HC (r = -.29; z = -2.29, p < .05). Table 2 shows the finding of r = -.29 for the TE group and r = -.20 for the HC group.

7.     The finding pertaining to the MDI appears to have resulted from a large number of exploratory analyses (i.e., unjustified examination of correlations with measure subscales – giving the appearance of “fishing”.) This raises concerns about type I error. The finding is interesting but requires some type of caveat or statistical correction.

8.     On p. 6, it is stated incorrectly that participants with PTSD “evinced a higher blink rate … in the face of threatening images.” Though it is true that they showed this effect for threatening IAPS images, they did NOT show this effect for threatening faces (which are also a type of image). Furthermore, they showed a near significant effect for non-threatening IAPS images as well. Thus the authors need to explain why the PTSD participants were more sensitive to IAPS vs. facial images. “Threat” may not be the most salient issue here.

9.     On pg. 6, clarify the phrase “… EBR predicts success of processes responsible for inhibiting task-irrelevant stimuli.” One or more words appears to be missing. What exactly is inhibited? The stimuli are not inhibited. Are they referring to memories or attention that is inhibited or something else?

10.  The authors refer (on pg. 6, line 210) to “negative correlations between EBR and flanker interference” to bolster their “cognitive control” theory. However, these negative correlations are non-significant and thus cannot be presumed to be different from zero.  

11.  On line 211, there is a period missing after “greater cognitive control.”

Author Response

POINT-BY-POINT RESPONSES TO REVIEWERS

Reviewer 2:

1a. EBR is described as “a broad index of information processing functions” (p. 2). This seems vague. Though space is limited, it would be helpful to have a concise description of the known neurophysiology of EBR to help the reader to understand its link with “information processing.”

 We agree with the reviewer that our original wording was imprecise.  In  the current version we have included a brief description of known neurophysiology of spontaneous eye blinks , as distinct from either voluntary or reflexive eye blinks. Neuroimaging analysis of spontaneous blinks indicate two main (non-ocular) regions of increased cortical activity: the parahippocampal gyrus and the medial frontal cortex (BA 6; Yoon, Chung, Song, & Park, 2004).  The parahippocampal region has long been associated with the updating of working memory to task-relevant stimuli in tasks such as the active oddball task (Halgren, Baudena, Clarke, Heit, Liègeois, et al., 1995; Halgren et al., 1998; Knight, Scabini, Woods, & Clayworth, 1989).  The medial frontal cortex is thought to be a cognitive control center, which is activated in tasks involving the monitoring and resolution of conflict from task-irrelevant stimuli, as occurs in the flanker paradigm used in the current study (Ridderinkhof, Ullsperger, Crone, & Nieuwenhuis, 2004).  sEBR is thus especially appropriate in understanding cognitive dysfunction in PTSD, which is thought to involve compromise to both working memory and executive control systems.

1b. The information given about the dopaminergic underpinnings of EBR could suggest that it indexes motor function.

You make a very good point.  It seems that in linking sEBR to severe disease states such as Tourette’s syndrome or Parkinson’s disease we inadvertently gave the impression that blinks gauge disruption of motor function.  In fact, sEBR in non-disease states taps cognitive load independent of motor function (Siegle, Ichikawa & Steinhauer, 2008).  We have clarified this distinction in the current version.    

1c. Similarly, the authors state but do not explain links between EBR and “inhibitory control” in prior literature. Indeed the reader is told to expect that EBR would predict poorer inhibitory control in the PTSD group but better inhibitory control in the control groups.

We have extended the discussion of how interference and inhibitory control are terms that describe the same cognitive process. While there were links in the paper, we realize that the different terminology may have made it unclear. We sought in this version to clarify our predictions regarding the relationship between sEBR and attentional interference in PTSD and TE individuals.  In the case of TE, we expected to find, as in previous research with healthy controls, that increases in sEBR are associated with better inhibitory control (i.e., less attentional interference in the flanker task).  By contrast, in the case of PTSD, we expected the converse relationship: As sEBR increases, attentional inteference also increases.  Our expectations derive from differences in how the two groups process threat material in working memory: In PTSD, threat material accumulates with increased sEBR, whereas in TE threat material dissipates with increased sEBR.  We hope our predictions are clearer in the current version.

1d. We are told that EBR reflects “accumulation of threat material in working memory” in the PTSD group but “adaptive unburdening of emotional material from working memory” in the control groups. This is confusing!  

We agree that the original phrasing, though precise, was difficult to parse.  In the current version we clarify that in PTSD EBR reflects the filling up of working memory with threat material whereas in the control groups EBR reflects the emptying out of threat material from working memory.

2.     Explain why participants are “asked to blink before or after, but not during, trials as needed.” Does the study aim to partly evaluate the participant’s ability to control blinking in the face of threatening stimuli? If so, this could be more clearly stated.  In any case, this instruction raises concerns about characterizing the eyeblinks assessed in this study as “spontaneous.”

The larger study probed ERP responses to threatening and neutral stimuli.  Thus, asking participants to avoid making blinks during stimulus presentation is a standard instruction to prevent artifacts during EEG recording. In the text, we have now clarified the instructions and our rationale. We agree with the reviewer that our instructions raise a possible concern over how spontaneous was our sEBR.  However, “spontaneous” blinks refer primarily to the type and rate of blinks – it is unusual for participants to blink during the brief (1 second) periods of stimulus presentation used in this experiment.  Studies that evaluate blink control normally use much longer stimulus presentation latencies. We feel confident, therefore, that blinks during our tasks were largely spontaneous in nature.  We also demonstrate that these data meet the criteria for spontaneous eye blinks with characteristic electrophysiological patterns (namely amplitude of the EOG measurement).

3.     Inhibitory control could be better defined.  Explain why more flanker interference indicates more inhibitory control to convince the reader that one can be substituted for the other.

In response to the reviewer, we now define more fully the different forms of inhibitory control and the relationship between inhibitory control and flanker interference.  Inhibitory control is an executive control process that actively suppresses task-irrelevant memories and experiences.  The avoidance of interference in the flanker task represents a specific type of inhibitory control (inhibitory interference control, as contrasted with inhibitory action control) because inhibition suppresses and helps resolve stimulus conflict from the distracting flanker. We sought in this version to underscore the point that, at least in healthy controls, greater inhibitory control is normally associated with less flanker interference, not more.  We agree that this point could have been clearer and have added additional description to elucidate it.

4.     Justify the interest in contrasting faces and non-faces in the study design. The results from the two types of threat cues are not consistent.

 We included two types of threat cues – faces and IAPS images – in the current study to evaluate the generality of the relationship between sEBR and flanker interference in PTSD when confronted with threatening versus non-threatening material.  Research from our laboratory indicates that individuals with PTSD reveal abnormally high levels of trust to faces typically viewed as untrustworthy (Fertuck et al., 2016).  Threatening images from the IAPS collection are commonly deployed in studies of threat sensitivity in PTSD.  However, few studies have examined responses to these different stimulus sets in the same sample of PTSD patients.  Our results here reveal differences between the two sets of stimuli because the relationship between flanker interference and EBR was statistically significant only in the IAPS set.  However, the pattern of correlations – a positive relationship between interference and EBR in PTSD and a negative relationship in TE and HC – was very consistent across the two sets of  stimuli.

5.     Table 1 would be improved by comparing the groups on mean levels of all dependent measures (i.e., EBR and flanker interference) for each stimulus condition. The interpretation of the correlations would be aided by knowing whether, for example, the PTSD group had a higher overall EBR (or only higher EBR under certain conditons).

 We have included the means and standard deviations for both EBR and flanker interference.

6.     There is a discrepancy between the description of some results in the text (pg. 5) and Table 2. Specifically, the text reports “negative correlations between EBR and flanker interference in the TE (r = -.20; z = -2.29, p < .05) and HC (r = -.29; z = -2.29, p < .05). Table 2 shows the finding of r = -.29 for the TE group and r = -.20 for the HC group.

  Thank you for pointing out this inconsistency. We have corrected the typo in the table.

7.     The finding pertaining to the MDI appears to have resulted from a large number of exploratory analyses (i.e., unjustified examination of correlations with measure subscales – giving the appearance of “fishing”.) This raises concerns about type I error. The finding is interesting but requires some type of caveat or statistical correction.

We agree.  In this version we have clarified that no apriori hypotheses were formulated for these analyses and therefore we consider them exploratory and evaluate the findings in this light.

8. On p. 6, it is stated incorrectly that participants with PTSD “evinced a higher blink rate … in the face of threatening images.” Though it is true that they showed this effect for threatening IAPS images, they did NOT show this effect for threatening faces (which are also a type of image). Furthermore, they showed a near significant effect for non-threatening IAPS images as well. Thus the authors need to explain why the PTSD participants were more sensitive to IAPS vs. facial images. “Threat” may not be the most salient issue here.

These are excellent points.  We do not have a ready explanation for why the correlations between EBR and flanker interference are slightly higher to IAPS images versus faces.  We do note that the direction of correlation is consistent within groups.  Of concern, however, is your point that both threat and non-threat images yield correlation coefficients of nearly equal magnitude in each group.  This finding obviously places limits on the emotional management account that we put forth in the Discussion.  In the current version we have tempered our discussion of these findings.

9.     On pg. 6, clarify the phrase “… EBR predicts success of processes responsible for inhibiting task-irrelevant stimuli.” One or more words appears to be missing. What exactly is inhibited? The stimuli are not inhibited. Are they referring to memories or attention that is inhibited or something else?

 We have clarified the sentence as follows: “Hester and Garavan [17] hypothesized that sEBR predicts the success of cognitive processes responsible for inhibiting the influence of task-irrelevant stimuli on performance.” In stimulus conflict situations, such as the flanker task, inhibitory processes are needed to lessen the influence of conflicting information on decision making (i.e., deciding on the basis of the flanker’s identity rather than on the basis of the target’s identify). Hester and Garavan suggest that sEBR gauges the success of these inhibitory processes, perhaps through an increased activation of medial frontal control regions.  

10.  The authors refer (on pg. 6, line 210) to “negative correlations between EBR and flanker interference” to bolster their “cognitive control” theory. However, these negative correlations are non-significant and thus cannot be presumed to be different from zero.  

We agree, and have indicated only that the correlations of the TEHC and HC groups were not significant.

11.  On line 211, there is a period missing after “greater cognitive control.”

We have added the missing period.

Round 2

Reviewer 2 Report

Title: Inhibitory control under threat: The role of spontaneous eye blinks in post-traumatic stress disorder

Summary of Manuscript. This a first revision of a manuscript describing a study comparing participants with posttraumatic stress disorder (PTSD; n = 19), trauma-exposed controls without PTSD (TE; n = 16), and non trauma-exposed “healthy controls” without PTSD (HC; n = 15) the association between their spontaneous blink rate and extent of flanker interference.  The key finding is that the PTSD group showed a trend toward overall positively signed correlations between EBR and flanker interference which significantly differed from the seemingly negative correlations observed in the control groups. Only one of the 12 reported correlations was statistically significant, i.e., PTSD subjects showed a moderately strong positive association (r = .51) between EBR and flanker interference when presented with threat-related non-face (IAPS) slides. This correlation was significantly different from what was observed in the other two participant groups. The authors also noted a significant negative correlation between EBR and a psychometic subscale indexing emotional constriction (r = -.48). However, this correlation was observed after a large series of exploratory analyses. They conclude that “EBR represents a potential physiological index of emotional management in PTSD.”

Overall Impression. The authors were thoughtful and responsive to my prior detailed feedback. The manuscript is definitely improved in many ways. I found it informative and derived more pleasure from reading it.

Major Remaining Concerns:

1.     The authors continue to interpret their findings in light of a theory that “threat material accumulates with increased sEBR in PTSD, whereas in TE threat material dissipates with increased sEBR.” For example, on pg. 7 they state “Our findings indicate that individuals in these two groups were able to release threatening and neutral material equally well from working memory.” The claim of “release of material” seems speculative to me and not rooted in any specific empirical data from their study. I find their speculation plausible but not on solid empirical ground. Thus I strongly recommend that they clearly label this idea as speculation in need of direct study throughout the manuscript (not only in the Discussion section). They should also use more tentative language in making this line of interpretation. For example, on the bottom of pg. 7, they should write, “In particular, sEBR in this population COULD GAGUE the buildup of threat material within working memory…”

2.     The authors continue to minimize the fact that the IAPS images seemed to be more potent elicitors of their findings than the face cues. I found the final sentence of the Discussion very unsatisfying in this regard and recommend further thoughht. It is striking that the “threat faces” seem to produce the WEAKEST correlation between sEBR and flanker interference in the PTSD group. Though probably not significantly different from the other correlations in this group, the pattern of results points to a more general deficit that is particularly provoked by the IAPS threat cues (the authors now hint at this in their discussion). Greater exploration of what distinguishes the IAPS cues from the face cues would be welcome or at least raised as a topic for further inquiry. I also wonder if the findings are at all explainable by considering the (right) lateral brain specialization for processing faces. 

3.     The Conclusion incorrectly implies that the control groups showed “suppressed levels of flanker interference” and the PTSD group showed “elevated” flanker interference. Table 1 shows that this is not true. The three groups did not differ in flanker interference. The authors meant to state that the PTSD group showed a positive association between sEBR and flanker interference that was significantly different from the seemingly negative association between these two variables in the control groups. It is important to describe these findings as precisely (and accurately) as possible, especially in the Conclusion, which may the only part of the paper read by some busy readers.

4.     If space allows, the abstract should include a sentence or two stating the major limitations of the study. For example, I feel pretty strongly that the authors must point out that their data do not support the interpretation that “threat” cues uniquely elicit positive correlations between sEBR and flanker interference. This is an unexpected result that should not be hidden from casual readers. They should also acknowledge that most correlations were not statistically significant and warrant replication in a larger sample.   

Other Minor Concerns:

1.     A key sentence at the bottom of pg. 1 is long and confusing. I recommend modifying it as follows: “The aim of the current study was to investigate the relationship between working memory (as indexed by spontaneous eye blink rate; sEBR) and inhibitory control of attentional interference (as measured by behavioral performance on the temporal flanker paradigm) in participants: (a) with PTSD, (b) with trauma exposure but no PTSD, and (c) with neither trauma exposure nor PTSD.”

2.     On the top of page 2, the phrase “conflict-induced” is unexpected and thus confusing. Explain or omit.

3.     On pg. 2, the phrase “may instigate the inhibitory control to threat observed in PTSD” is imprecise. Should this be interpreted as “REDUCED inhibitory control to threat…”? If so, please clarify.

4.     I appreciated the helpful clarification on pg. 4 about why participants were asked not to blink during the stimulus presentation. However, it would help further to describe the specific time window in which the spontaneous eyeblinks were measured relative to the times when they were instructed not to blink. The response letter gave me the impression that the stimulus presentation window was very small relative to the window in which blinks were assessed. If so, please share this information with the readers.

5.     I appreciated the addition of descriptive data in Table 1. However, the main manuscript text in the Results section should note that the groups did not differ in mean sEBR to any condition or Flanker interference. Also, the Table note should explain the IAPS abbreviation.

6.     If the exploratory correlations are going to remain in the paper (pg.7), which I leave up to editorial discretion, the authors should clearly state the number of correlations they conducted to produce the one significant finding. In other words, they should state how many subscales were crossed with the overall blink rate scores in each of the three groups.  

7.     On pg. 7, the phrase “at most, they exhibited the usual relationship…” is misleading in light of the non-significant results. I would prefer rephrasing to state “at most, they exhibited a trend toward the usual relationship…”

Author Response

Response to Reviewer 2: Round 2

1.     The authors continue to interpret their findings in light of a theory that “threat material accumulates with increased sEBR in PTSD, whereas in TE threat material dissipates with increased sEBR.” For example, on pg. 7 they state “Our findings indicate that individuals in these two groups were able to release threatening and neutral material equally well from working memory.” The claim of “release of material” seems speculative to me and not rooted in any specific empirical data from their study. I find their speculation plausible but not on solid empirical ground. Thus I strongly recommend that they clearly label this idea as speculation in need of direct study throughout the manuscript (not only in the Discussion section). They should also use more tentative language in making this line of interpretation. For example, on the bottom of pg. 7, they should write, “In particular, sEBR in this population COULD GAGUE the buildup of threat material within working memory…”

We appreciate the reviewer’s recommendation.  In the current version we have reworked the language to clarify the difference between speculation and claims backed by empirical data. In the Introduction, we describe as a “speculative hypothesis” the account proposed of sEBR by Ichikawa and Ohira (2004) as involving the release of representations from working memory information. Speculation of ours, relegated to the Discussion, is now clearly labeled as such: “We speculate that there may be an emotional component to this process”; “Individuals in these two groups may be able to release threatening and neutral material..”; “flanker interference found in the current study may represent poor overall inhibitory control in PTSD”; “sEBR in this population could gauge the buildup of threat material within working memory”; “To be sure, at present our explanations are speculative and our results in need of replication.”

2.     The authors continue to minimize the fact that the IAPS images seemed to be more potent elicitors of their findings than the face cues. I found the final sentence of the Discussion very unsatisfying in this regard and recommend further thought. It is striking that the “threat faces” seem to produce the WEAKEST correlation between sEBR and flanker interference in the PTSD group. Though probably not significantly different from the other correlations in this group, the pattern of results points to a more general deficit that is particularly provoked by the IAPS threat cues (the authors now hint at this in their discussion). Greater exploration of what distinguishes the IAPS cues from the face cues would be welcome or at least raised as a topic for further inquiry. I also wonder if the findings are at all explainable by considering the (right) lateral brain specialization for processing faces. 

 As the reviewer encouraged, we have given some thought to the implications of this finding.  What we found striking is that the difference in correlations between stimulus sets was restricted to the PTSD group.  This suggests that the effect may be due to the differential impact of threatening stimuli on PTSD patients.  We offer the following tentative explanation: “In the PTSD group the IAPS stimuli yielded numerically higher correlation coefficients than the face stimuli (see Table 2).  Interestingly, the difference between stimulus sets was particularly robust to threat stimuli.  The effect cannot be understood merely as weaker responding in PTSD from facial processing because faces also appeared in many of the IAPS images.  Perhaps IAPS images are more potent stimuli for eliciting threat because, in depicting threatening scenes, they are more directly threatening to PTSD participants, whereas, in expressing fear, NimStim faces depict only the aftermath of (i.e., the response to) threat.” We look forward to the reviewer’s input regarding this explanation.

3.     The Conclusion incorrectly implies that the control groups showed “suppressed levels of flanker interference” and the PTSD group showed “elevated” flanker interference. Table 1 shows that this is not true. The three groups did not differ in flanker interference. The authors meant to state that the PTSD group showed a positive association between sEBR and flanker interference that was significantly different from the seemingly negative association between these two variables in the control groups. It is important to describe these findings as precisely (and accurately) as possible, especially in the Conclusion, which may the only part of the paper read by some busy readers.

 We agree and have worked to clarify that it is the direction of the associations, not their magnitude, that is different between groups: We showed that whereas increasing rates of spontaneous eye blinks are associated negatively with flanker interference in TE and HC, sEBR and flanker interference are actually associated positively in PTSD.”

4.     If space allows, the abstract should include a sentence or two stating the major limitations of the study. For example, I feel pretty strongly that the authors must point out that their data do not support the interpretation that “threat” cues uniquely elicit positive correlations between sEBR and flanker interference. This is an unexpected result that should not be hidden from casual readers. They should also acknowledge that most correlations were not statistically significant and warrant replication in a larger sample.   

 We agree and have included a sentence in the abstract highlighting the discrepancy between the hypotheses and results: “Although our results are suggestive of sEBR as a potential physiological index of emotional management in PTSD, most of the correlations fell below significance to a trend level, indicating that further research with a larger sample is needed.”

Other Minor Concerns:

1.     A key sentence at the bottom of pg. 1 is long and confusing. I recommend modifying it as follows: “The aim of the current study was to investigate the relationship between working memory (as indexed by spontaneous eye blink rate; sEBR) and inhibitory control of attentional interference (as measured by behavioral performance on the temporal flanker paradigm) in participants: (a) with PTSD, (b) with trauma exposure but no PTSD, and (c) with neither trauma exposure nor PTSD.”

 We appreciate the revision provided and have modified the text as recommended.

2.     On the top of page 2, the phrase “conflict-induced” is unexpected and thus confusing. Explain or omit.

 We have omitted the phrase “conflict-induced” and replaced it with “attentional”, which is clearer in the current context.

3.     On pg. 2, the phrase “may instigate the inhibitory control to threat observed in PTSD” is imprecise. Should this be interpreted as “REDUCED inhibitory control to threat…”? If so, please clarify.

 Thank you for pointing out the imprecision.  We have clarified that it is indeed reduced inhibitory control to threat.

4.     I appreciated the helpful clarification on pg. 4 about why participants were asked not to blink during the stimulus presentation. However, it would help further to describe the specific time window in which the spontaneous eyeblinks were measured relative to the times when they were instructed not to blink. The response letter gave me the impression that the stimulus presentation window was very small relative to the window in which blinks were assessed. If so, please share this information with the readers.

 We have included additional information regarding stimulus duration within the overall duration of the trial: “The duration of stimuli was 1265 ms on average within a trial that lasted approximately 3665 ms.”

5.     I appreciated the addition of descriptive data in Table 1. However, the main manuscript text in the Results section should note that the groups did not differ in mean sEBR to any condition or Flanker interference. Also, the Table note should explain the IAPS abbreviation.

 We have noted that the groups did not differ in mean sEBR or flanker interference. We have explained the IAPS abbreviation.

6.     If the exploratory correlations are going to remain in the paper (pg.7), which I leave up to editorial discretion, the authors should clearly state the number of correlations they conducted to produce the one significant finding. In other words, they should state how many subscales were crossed with the overall blink rate scores in each of the three groups.  

We now state the number of correlations computed during the exploratory analyses by specifying the number of subscales included for each clinical measure.  Despite finding only a single significant correlation, we elected to retain the exploratory analyses to aid our understanding in the Discussion of the full pattern of results across groups.  We agree that the final decision regarding the fate of these analyses rests with the editors.

 We agree that in light of the number of comparisons we conducted it is more appropriate to remove this finding as it is unlikely to be better than chance that it emerged as significant.

7.     On pg. 7, the phrase “at most, they exhibited the usual relationship…” is misleading in light of the non-significant results. I would prefer rephrasing to state “at most, they exhibited a trend toward the usual relationship…”

We agree and have amended the phrase to include the statement that it is a trend.

Round 3

Reviewer 2 Report

Title: Inhibitory control under threat: The role of spontaneous eye blinks in post-traumatic stress disorder

Summary of Manuscript. This a second revision of a manuscript describing a study comparing participants with posttraumatic stress disorder (PTSD; n = 19), trauma-exposed controls without PTSD (TE; n = 16), and non trauma-exposed “healthy controls” without PTSD (HC; n = 15) on the association between their spontaneous blink rate and extent of flanker interference.  The key finding is that the PTSD group tended to show overall positively signed correlations between EBR and flanker interference which significantly differed from the seemingly negative correlations observed in the control groups. Only one of the 12 reported correlations was statistically significant, i.e., PTSD subjects showed a moderately strong positive association (r = .51) between EBR and flanker interference when presented with threat-related non-face (IAPS) slides. This correlation was significantly different from what was observed in the other two participant groups. After exploratory analyses, the authors also found a significant negative correlation between EBR and a psychometic subscale indexing emotional constriction (r = -.48).

Overall Impression. Like before, the authors were responsive to my prior detailed feedback. The manuscript continues to be improved especially with respect to adopting a more circumspect tone. The remaining major concern is that the authors don’t seem to have a clear explanation for their results. They now speculate that “Perhaps IAPS images are more potent stimuli for eliciting threat because, in depicting threatening scenes, they are more directly threatening to PTSD participants, whereas in expressing fear, … faces depict only the aftermath of (i.e., the response to) threat.” I think that they are probably on the right track now. It seems to me that their data shows that the “fear faces” were never really experienced as “threatening” to the PTSD group. It would be helpful if the authors had some other data (e.g., subjective ratings) to argue that the face stimuli were comparable to the IAPS stimuli. In the absence of such data I think that they are on more solid ground to suggest that the IAPS scenes were experienced as potential trauma cues while the face stimuli were not. It might have been different if they used, for example, angry faces that could have been reminiscent of perpetrators of the physical abuse and sexual trauma experienced by the majority of their participants. Having said that, they are still left with the problem of explaining why the “non-threat IAPS” images yielded similar effects to the “threat IAPS” images. I think that this may be explained by the fact that images were presented in a random order. Once PTSD patients encounter a trauma cue they can have difficulty inhibiting a threat response to a “safe” cue especially when there is some ambiguity (which may be true of the non-threat IAPS images). In other words, PTSD participants may be able to easily distinguish the faces from IAPS but have more difficulty distinguishing threat IAPS from non-threat IAPS (or at least difficulty inhibiting their anxiety about the images). See Pole, Neylan, Best, Orr, & Marmar (2003). Fear-potentiated startle and posttraumatic stress symptoms in urban police officers. Journal of Traumatic Stress, 16, 471-479  for a similar result in a different paradigm. 

I have one additional request. I think that it would be helpful if the authors listed all of the items comprising the “emotional constriction” scale in the Discussion so that the readers can have more information about the abnormality that seems to be associated with EBR. This may illustrate that the PTSD patients who show elevated EBR and elevated flanker interference in response to the IAPS slides also have trouble inhibiting negative emotion, which could explain why seeing any threatening IAPS slides contaminates their response to all other stimuli esp other IAPS slides. 

Author Response

Response to Reviewer 1: Round 3

 1. Like before, the authors were responsive to my prior detailed feedback. The manuscript continues to be improved especially with respect to adopting a more circumspect tone. The remaining major concern is that the authors don’t seem to have a clear explanation for their results. They now speculate that “Perhaps IAPS images are more potent stimuli for eliciting threat because, in depicting threatening scenes, they are more directly threatening to PTSD participants, whereas in expressing fear, … faces depict only the aftermath of (i.e., the response to) threat.” I think that they are probably on the right track now. It seems to me that their data shows that the “fear faces” were never really experienced as “threatening” to the PTSD group. It would be helpful if the authors had some other data (e.g., subjective ratings) to argue that the face stimuli were comparable to the IAPS stimuli. In the absence of such data I think that they are on more solid ground to suggest that the IAPS scenes were experienced as potential trauma cues while the face stimuli were not. It might have been different if they used, for example, angry faces that could have been reminiscent of perpetrators of the physical abuse and sexual trauma experienced by the majority of their participants.

In seeking to understand responses to threat in PTSD we relied on emotional stimuli used widely in the literature probing the neurocircuitry of fear and anxiety.  Stimuli depicting threatening scenes, including the IAPS images we employed, are common (Britton et al, 2006; Hariri et al, 2002; Irwin et al, 1996; Lane et al, 1997; Paradiso et al, 1999; Phan et al, 2003; Reiman et al, 1997; Taylor et al, 1998).  Interestingly, stimuli depicting fearful faces are usually equally effective in activating the fear/anxiety neurocircuitry (Breiter et al, 1996; Davis and Whalen, 2001; Fitzgerald et al, 2006; Gorno-Tempini et al, 2001; Morris et al, 1996; Phillips et al, 1997, 2004; Sabatini et al, 2009; Vuilleumier and Pourtois, 2007; Whalen et al, 2001; see Shin & Liberzon, 2010, for a review).  Stimuli depicting angry faces may have been more potent than fearful faces in eliciting a threat response in our cue-reactivity paradigm, but we elected to employ the latter because of their prevalence and effectiveness in the literature. We agree with the reviewer that an inevitable conclusion to our findings and their interpretation is that the fear faces were not, or were only weakly, experienced as trauma cues.  We make this conclusion in the current version.

2. Having said that, they are still left with the problem of explaining why the “non-threat IAPS” images yielded similar effects to the “threat IAPS” images. I think that this may be explained by the fact that images were presented in a random order. Once PTSD patients encounter a trauma cue they can have difficulty inhibiting a threat response to a “safe” cue especially when there is some ambiguity (which may be true of the non-threat IAPS images). In other words, PTSD participants may be able to easily distinguish the faces from IAPS but have more difficulty distinguishing threat IAPS from non-threat IAPS (or at least difficulty inhibiting their anxiety about the images). See Pole, Neylan, Best, Orr, & Marmar (2003). Fear-potentiated startle and posttraumatic stress symptoms in urban police officers. Journal of Traumatic Stress, 16, 471-479  for a similar result in a different paradigm. 

We thank the reviewer for the suggestions in how best to understand the absence of a difference in correlation between threat and non-threat conditions. The article that the reviewer recommended (Pole et al. 2003), and also Grillon and Morgan (1999), reported a conceptually similar finding to ours, namely, behavioral or physiological responsivity in PTSD participants under conditions of low or no threat.  These authors interpreted their results as reflecting the manner in which PTSD patients appraise environmental risk: by vigilantly monitoring both safe and dangerous scenarios. We believe that such an explanation may also apply to blink responses, which we suggest in the manuscript emerges from the emotional management of threatening (and nonthreatening) material in working memory.  Such an account emphasizes the contextual nature of risk appraisal in PTSD.  We include this explanation in the current revision.

3. I have one additional request. I think that it would be helpful if the authors listed all of the items comprising the “emotional constriction” scale in the Discussion so that the readers can have more information about the abnormality that seems to be associated with EBR. This may illustrate that the PTSD patients who show elevated EBR and elevated flanker interference in response to the IAPS slides also have trouble inhibiting negative emotion, which could explain why seeing any threatening IAPS slides contaminates their response to all other stimuli esp other IAPS slides. 

We agree and have provided a footnote where we indicate the five items of the “emotional constriction” scale.
